# Collaborative Ring Trial of the Applicability of a Reference Plasmid DNA Calibrant in the Quantitative Analysis of GM Maize Event MON810

**DOI:** 10.3390/foods11111538

**Published:** 2022-05-24

**Authors:** Yanan Meng, Shu Wang, Jinchao Guo, Litao Yang

**Affiliations:** 1Pilot National Laboratory for Marine Science and Technology, Qingdao 266000, China; ynmeng@qnlm.ac; 2National Center for the Molecular Characterization of Genetically Modified Organisms, Joint International Research Laboratory of Metabolic and Developmental Sciences, School of Life Sciences and Biotechnology, Shanghai Jiao Tong University, Shanghai 200240, China; wangshu@sjtu.edu.cn (S.W.); jcwguo@sjtu.edu.cn (J.G.)

**Keywords:** plasmid-DNA-based reference material, collaborative ring trial, statistical analysis, pMON810, quantitative real-time PCR

## Abstract

Certified reference materials (CRMs) is one of the critical requirements in a quantitative analytical method, such as in the quantification of genetically modified (GM) contents in food/feed products. Plasmid-DNA-based CRMs are becoming essential in GM content quantification. Herein, we report the construction of one plasmid DNA calibrant, pMON810, for the quantification of the GM maize event MON810 which is commercially planted and used for food/feeds worldwide, and the collaborative ring trial was used to validate its applicability. pMON10 was proven to have high specificity for the MON810 event. The limit of detection (LOD) and limit of quantification (LOQ) of real-time PCR assays of MON810 event and maize endogenous gene using pMON810 as calibrant was 2 copies/μL and 5 copies/μL, respectively. A total of eight laboratories participated in the ring trial and returned valid test results. Each sample was performed with three repeats and three parallels in each repeat. Statistical analysis of the ring trial results showed that pMON810 as a calibrant had high PCR efficiency (ranging from 0.885 to 1.008) and good linearity (ranging from 0.9933 to 0.9997) in MON810 and endogenous gene real-time PCR assays. The bias between the test values and true values ranged from 4.60 to 20.00% in the quantification of five blind samples. These results indicate that pMON810 is suitable for use as a calibrant for the quantification of MON810 events in routine lab analysis or to evaluate detection methods for MON810, as well as being used as a substitute for the matrix-based CRM of MON810.

## 1. Introduction

Since 1994, GM crops have been developed and planted worldwide. By the end of 2020, the global GM crop fields had reached 190.4 million hectares, increasing 112-fold compared with the number of 1.7 million hectares in 1996 [1]. GM maize is the most planted GM crop species with 60.9 million hectares in 244 various transgenic events, among which MON810 is the primary event [2,3]. Many countries and regions have issued and implemented genetically modified organism (GMO) labeling regulations to address public concerns about the safety issues of GM crops and products [4]. To enforce these GMO labeling regulations, it is important to develop accurate and flexible GMO detection methods and related reference materials (RMs). Currently, real-time quantitative PCR (qPCR) is the primarily used method for quantification of GM content in food/feed products, though many other new GMO detection techniques such as microarray, digital PCR, re-sequencing, and biosensor, etc., were also reported [5,6,7,8]. Quantitative PCR analyses of GM content can be grouped into four categories: screening-specific, gene-specific, construct-specific, and event-specific [9]. Among them, screening-specific qPCR targets commonly used transgenic elements and/or marker genes to detect the existence of GM content, and event-specific qPCR targets junction regions between host genomic DNA and exogenous DNA to identify specific events [10].

During quantitative qPCR analysis of GM contents, reference materials (RM) are necessary for maintaining the accuracy of laboratory measurements over time and inter-laboratory comparability of quantitative results [11,12]. Currently, various types (such as matrix-based, plasmid-DNA-based, genomic-DNA-based, and seed-based) of RMs have been developed for GMO detection in the EU, USA, Japan, and China [11,12]. At present, over 200 CRMs have been developed, containing 22 genomic-DNA-based CRMs, 188 matrix-based CRMs, and 9 plasmid-DNA-based CRMs [12]. This number of developed CRMs remains far less compared with the large number of commercialized GM crop events (more than 400), reflecting the reality that more CRMs need to be developed quickly [12,13]. Matrix-based CRMs remain the most widely used because they present similar properties to real samples and are easily traceable to the International System of Units (SI) of the gram [11,14]. The disadvantages of matrix-based CRMs in GMO quantitative analysis include the narrow dynamic range of quantification, multiple and complex preparation procedures, high cost, and difficulty in obtaining the homogeneous candidate [15,16]. Compared with matrix-based CRMs, plasmid-DNA-based CRMs are good substitutes in GMO analysis since they have a well-characterized transgene sequence and copy number, and they can be easily maintained in and produced from bacterium stocks [17,18,19,20,21]. In the detection of foodborne bacteria and viruses, plasmid-DNA-based CRMs are also widely used instead of inactivated bacteria and viruses [22,23].

The GM MON810 event produced by the Monsanto Company, USA, has been approved for planting or using as food/feed materials in many countries, such as the USA, EU, Japan, Korea, and China, etc. The development of RMs for the MON810 event is very important to ensure the legalized planting and production processing. During the development of a plasmid-DNA-based CRM, it is critical to validate its characteristic values in addition to its homogeneity and stability [24,25]. In this study, we constructed a plasmid DNA calibrant, pMON810, for quantitative qPCR analysis of GM maize MON810, and we organized a collaborative ring trial for inter-laboratory validation of the applicability and viability of pMON810 as a plasmid DNA calibrant.

## 2. Materials and Methods

### 2.1. Materials and DNA Extraction

Kernels of MON810 maize were generously supplied by the Monsanto Company. Seeds of wild-type maize were purchased from a market in Shanghai, China and certified as GMO- free by qPCR analysis in our lab. In accordance with the manual, maize genomic DNAs were isolated using the Plant DNA Mini-Prep Kit (Ruifeng Agro-tech Co. Ltd., Shanghai, China). Plasmid DNAs were isolated using the Plasmid Mini Extraction Kit (Qiagen, Hilden, Germany). The quality and quantity of extracted genomic or plasmid DNA were evaluated by measuring their UV absorption at 260 nm with a ND-2000 spectrophotometer (NanoDrop Technologies Inc., Rockland, DE, USA). The concentration of each extracted genomic or plasmid DNA was further measured using the Quant-IT PicoGreen dsDNA Assay Kit (Thermo Fisher Scientific, Shanghai, China). The copy number of extracted genomic or plasmid DNA was calculated according to the DNA amount and the size of the maize genomic DNA. The extracted plasmid DNA solution of pMON810 was adjusted to 10^7^ copies/μL for further experiments. For determining the conversion factor (Cf), a total of 5 MON810 maize genomic DNA samples with concentrations corresponding to 20, 4, 0.8, 0.16 and 0.032 ng/μL, were prepared and coded C1–C5. A total of 5 blind samples were prepared by mixing GM maize powder with non-GM maize powder directly in three different GM contents (mass ratio; 5.0, 1.0 and 0.5%) and coded X1–X5. After receiving the blind samples, each participant in the ring trial extracted and purified the genomic DNA and then diluted each DNA extraction to 20 ng/μL for further quantitative analysis. Salmon sperm DNA (20 ng/μL) was prepared and used as a negative DNA control for qPCR analysis.

### 2.2. Oligonucleotide Primers and TaqMan Probes

In this study, all oligonucleotide primers and TaqMan probes were designed with beacon designer version 8.0 (PREMIER Biosoft Ltd., San Francisco, CA, USA). The sequence information of the designed primers and probes was listed in Appendix A. The primer set con-MON810-1F/2R was used to construct the pMON810 plasmid. Primer set Q-MON810-1F/2R and probe Q-MON810-P were designed according to the 5’ event-specific sequence of MON810 and used for quantifying the amount of MON810 events. The primer set Q-*zSSIIb*-1F/2R and probe Q-*zSSIIb*-P targeting maize endogenous reference gene *zSSIIb* were used to quantify the total amount of maize genomic DNA. All primers and probes were purchased from Invitrogen Co. Ltd. (Shanghai, China).

### 2.3. Construction of pMON810

Plasmid pMON810 was constructed by introducing the event-specific maize DNA fragment into the plasmid pMaize [15]. The plasmid pMaize has a partial maize *zSSIIb* gene and a 2463-bp DNA fragment from the rice genome in tandem into a pBSK plasmid vector. The 892 bp event-specific sequence of MON810 was amplified and inserted between the *EcoR*I and *BamH*I restriction sites of pMaize. The detailed information of the inserted sequence was shown in Figure 1.

The PCR reaction to amplify the event-specific DNA fragment of MON810 was carried out in a Veriti Thermal Cycler (Thermo Fisher Scientific, Waltham, MA, USA) in a reaction volume of 50 μL, containing 1× PCR buffer with MgSO_4_, 0.2 mM dNTPs, 0.5 μM of each primer, 2.0 units of KOD-Plus DNA Polymerase (Toyobo Co., Kita-ku, Osaka, Japan), 2 μL MON810 genomic DNA (20 ng/μL), and brought to a final volume using ddH_2_O. The PCR program was as follows: pre-denaturation at 95 °C for 5 min, 35 cycles of 30 s at 95 °C, 60 s at 68 °C, and a final extension step at 72 °C for 5 min.

### 2.4. Real-Time PCR

The volume of each real-time PCR reaction was 25 μL, including 5 μL of template DNA, 12.5 μL 2× qPCR master mix (Zhejiang Yuzhi Biotechnology Company, Ningbo, China), 0.1 μM of each primer and 0.2 μM of probe, and brought to a final volume using ddH_2_O. All reactions were repeated three times, and three parallels per time. The real-time PCR procedure was carried out as follows: 95 °C for 10 min, 45 cycles of 95 °C for 15 s, and 60 °C for 60 s. Fluorescent signals were monitored at the annealing and extension step (60 °C) during each cycle. Participants in the inter-laboratory validation ring trial performed PCR amplifications using thermal cyclers in their own lab, including Rotor-Gene Q (Qiagen, Hilden, Germany), Prism ABI 7300, Prism ABI 7500, Prism ABI 7900 (Applied Biosystems, Foster City, CA, USA), and CFX96 Touch (Bio Rad, Hercules, CA, USA).

### 2.5. Collaborative Ring Trial

We invited eight laboratories focusing on GMO detection to join the collaborative ring trial of pMON810 applicability evaluation according to the internationally accepted guidelines [24,26]. Each participant received the following test items: one tube of plasmid pMON810 calibrant (10^7^ copies/μL, 100μL per tube), one tube of MON810 maize genomic DNA (20 ng/μL, 150 μL), 15 blind samples at five different GM content percentages in dried powder (coded X1-1, X1-2, X1-3 at 5.0% GM content; X2-1, X2-2, X2-3 at 1.0% GM content; X3-1, X3-2, X3-3 at 1.0% GM content; X4-1, X4-2, X4-3 at 0.5% GM content; X5-1, X5-2, X5-3 at 0.5% GM content. A total of 1 g per tube), 2 × qPCR master mix (1.8 mL × 3 tubes), and 5 mL DNA dilute buffer (1 mM ethylenediaminetetraacetic acid (EDTA), 10 mM Tris-HCl, pH 8.0). A single hard copy of the ring trial protocol and data report forms were also provided. All materials for the ring trial were transported at a low temperature through DHL International GmbH, Shanghai, China.

For the collaborative ring trial, each participant was requested to complete the validation according to the provided protocols. Firstly, each participant should prepare gradient diluted solutions with the concentrations of 10^6^, 10^5^, 10^4^, 10^3^, and 10^2^ copies/μL of pMON810 plasmid DNA using the provided DNA stock and DNA dilution buffer. The five DNA dilutions were used in the qPCR assays to construct the amplification standard curve. Salmon sperm DNA was used as a negative template control (NTC). Secondly, each participant was required to evaluate the conversion factor (Cf) of pMON810 in the qPCR assay using the provided GM maize genomic DNA (20 ng/μL, 150 μL). The provided GM maize genomic DNA samples (20 ng/μL, 150 μL) were diluted into five concentrations (using the supplied DNA dilution buffer) of 20 ng/μL, 4 ng/μL, 0.8 ng/μL, 0.16 ng/μL, and 0.032 ng/μL. The five concentrations were quantified using the constructed amplification standard curve to calculate the Cf values using the formula Cf = copy number of MON810/copy number of *zSSIIb*.

Finally, the five blind DNA samples were analyzed using qPCR assays for MON810 events and *zSSIIb* genes, and the GM contents were calculated using the constructed standard curve and the Cf values were determined by the equation: GM content (%) = (copy number of MON810 × 100)/(copy number of *zSSIIb*×Cf).

### 2.6. Statistical Analysis

Statistical analyses were performed on the returned data from all participants to determine several key parameters, such as qPCR amplification efficiency, linearity of constructed standard curves, Cf values, relative standard deviation of repeatability (RSD^R^), relative standard deviation of reproducibility (RSD^r^), relative deviation and bias of the quantification results.

## 3. Results and Discussion

### 3.1. Quantification of the Event-Specific Sequence of MON810 and Maize Endogenous zSSIIb Gene in pMON810 using qPCR

The developed pMON810 plasmid is 6539 bp in length and contains an event-specific sequence of MON810 and a maize endogenous *zSSIIb* reference gene (as shown in Figure 1). The purified pMON810 plasmid DNA was diluted to 10^9^ copies/μL (6.83 ng/μL) based on the size of the plasmid and the average amount of DNA (1 pg = 965 Mb) [15].

The performance of pMON810 as a calibrant in the qPCR assay was first evaluated in our laboratory to determine its limit of detection (LOD), limit of quantification (LOQ), and to construct the amplification standard curve. A total of six pMON810 DNA solutions at concentrations of 2000, 200, 100, 20, 5, and 2 copies/μL were prepared and analyzed to evaluate the LOD and LOQ. Event-specific DNA and *zSSIIb* DNA could be detected in all six DNA solutions, but high bias values (>25%) were observed in analyses of the DNA at 2 copies/μL. The LOD of the qPCR assays using pMON810 was determined as 10 copies per reaction and the LOQ was 25 copies per reaction. The constructed qPCR assay standard curve used 2000, 200, 100, 20, and 5 copies/μL of pMON810 DNA and demonstrated high PCR efficiency, good linearity, good repeatability and reproducibility, and an acceptable dynamic range of quantification.

### 3.2. Inter-Laboratory Ring Trial of pMON810 as a Calibrant in MON810 Maize Quantification

All eight participating laboratories sent back validation results within one month of receiving the ring trial package. The returned data included Ct values for all qPCR reactions at the same fixed threshold value, the formula of constructed standard curves, PCR efficiency, and linear correlation (R^2^). All these data were further analyzed statistically using the Cochran’s test without exclusions.

### 3.3. qPCR Efficiency and Linearity

All returned qPCR results were within the ring trial guidelines’ accepted range, including qPCR efficiency, linearity, and bias [27]. The qPCR standard curves of MON810 event-specific DNA and *zSSIIb* DNA amplification from all eight participants were shown in Figure 2. For the MON810 event-specific DNA assay, the qPCR efficiency was from 0.929 to 1.008 among eight laboratories with a SD of 0.03 and a RSD of 3.12%. The R^2^ was from 0.998 to 0.9997 with a SD of 0.001 and a RSD of 0.093%. For the endogenous gene *zSSIIb* assay, the PCR efficiency ranged from 0.885 to 1.049 with a SD of 0.05 and a RSD of 5.07%. The R^2^ ranged from 0.9933 to 0.9997 with a SD of 0.002 and a RSD of 0.216% (Table 1). These results indicated that pMON810, as a calibrant for MON810, has a high qPCR efficiency and acceptable linearity to ensure high precision in the quantification of practical GM samples.

### 3.4. qPCR Repeatability and Reproducibility Using pMON810 Plasmid DNA

The repeatability and reproducibility of MON810 and *zSSIIb* assays using pMON810 as a calibrator were evaluated. The standard deviation of repeatability (SD^r^) and relative deviation of repeatability (RSD^r^) were calculated from the Ct values from all repeats in each real-time PCR assay. In the MON810 assay, the SD^r^ ranged from 0.07 to 0.92, and the RSD^r^ ranged from 0.29 to 2.90%. In the *zSSIIb* assay, the SD^r^ ranged from 0.06 to 0.72, and the RSD^r^ ranged from 0.19 to 2.55%. The standard deviation of repeatability (SD^R^) and relative deviation of repeatability (RSD^R^) was calculated based on the mean Ct values from each repeat in each assay. In the eMON810 assay, the reproducibility SD^R^ ranged from 0.09 to 0.84, and the RSD^R^ ranged from 0.02 to 2.44%. In the *zSSIIb* assay, the SD^R^ ranged from 0.05 to 0.79, and the RSD^R^ ranged from 0.14 to 2.47%.

These SD and RSD values for both repeatability and reproducibility were well within the acceptable criteria for validated qPCR methods according to the ENGL guidelines, indicating that pMON810 is suitable for use as a calibrant for the quantification of the GM maize event MON810.

### 3.5. Measurement of the Conversion Factors (Cfs)

Because of the different sequence characteristics between plasmid DNA and genomic DNA, their qPCR amplification curves, and Ct values are often varied even for the same copy number of plasmid DNA and genomic DNA. To minimize the amplification rate difference between using a plasmid DNA and a genomic DNA template, Cf should be determined for pMON810 and used for blind sample quantification [15]. In each participating laboratory of the ring trial, the Cf values were determined using five MON810 genomic DNA solutions at different concentrations (20, 4, 0.8, 0.16, 0.032 ng/μL), and the results from all laboratories are listed in Table 2. The mean Cf values ranged from 0.51 to 0.69 with the SD values ranging from 0.02 to 0.10 and RSD ranging from 3.63 to 17.78%. The mean Cf values from all participants were around 0.6, indicating that there is no obvious variation between test results produced by different operators.

### 3.6. Blind Sample Quantification

The quantified GM contents of the five blind samples (X1–X5) from the eight participants are shown in Table 3. The mean values of the GM contents of the five blind samples were 5.23%, 1.10%, 1.20%, 0.55% and 0.59%, respectively. The quantified GM contents and the relative deviation of blind samples in each participant lab were shown in Figure 3. All labs quantified the GM contents of blind samples with high accuracy except for two tests with a slightly larger relative deviation over 35%. The larger relative deviation came from the blind sample X5 with 0.5% GM content from two laboratories, which might be caused by the different DNA extraction methods used in the participated laboratories and the quantitative accuracy of the qPCR assays decreased when using low concentration samples. We also observed that the relative deviation increased as the GM contents value decreased. Statistical analysis was performed to calculate the repeatability, reproducibility, and quantification bias (as shown in Table 4) according to the guidelines for qPCR method validation [27]. In repeatability evaluation, the SD^r^ values ranged from 0.08 to 0.54, and the RSD^r^ values ranged from 8.64 to 16.28%. For reproducibility, the SD^R^ values ranged from 0.12 to 0.69, and the RSD^R^ values ranged from 12.49 to 21.94%. The good repeatability and reproducibility of this ring trial indicated that the pMON810 is reliable in qPCR analysis when used as a calibrant, even though the instruments and operators were different. The bias values of the tested average GM contents of the five blind samples were from 4.00 to 17.00%, which is obviously lower than the generally accepted value of 25% in an acceptable qPCR method [27]. These results indicated that pMON810 could be used as a reliable calibrant to quantify GM maize MON810 content in practical samples by qPCR with high accuracy.

### 3.7. Two-Factor ANOVA Analysis

The relative deviation of results among the eight participants of the inter-laboratory validation was also determined by fully non-crossed two-factor ANOVA (as shown in Table 5). In the ANOVA analysis, laboratories and tested GM content results from each laboratory were used as two experimental factors. No significant difference was observed on the quantification results from different laboratories, indicating that the qPCR assays employing pMON810 as a calibrant for MON810 had good applicability in various laboratory settings.

## 4. Conclusions

In this study, we constructed a plasmid pMON810 containing the event-specific sequence of MON810 and the maize endogenous reference gene *zSSIIb* and validated its applicability in quantitative PCR analysis of GM MON810 through a collaborative ring trial involving eight laboratories. The study results demonstrated that pMON810 has high PCR amplification efficiency, good repeatability, good reproducibility, and high accuracy. The ring trial results also supplied creditable data for further certifying the pMON810 as a CRM. All these results confirm that the pMON810 is a suitable alternative to matrix-based CRMs for the quantification of MON810 maize and its derivatives in routine lab analysis. In addition, the pMON810 could be used as a calibrator to evaluate the performance of a new method or equipment for GM MON810 analysis.

## Figures and Tables

**Figure 1 foods-11-01538-f001:**
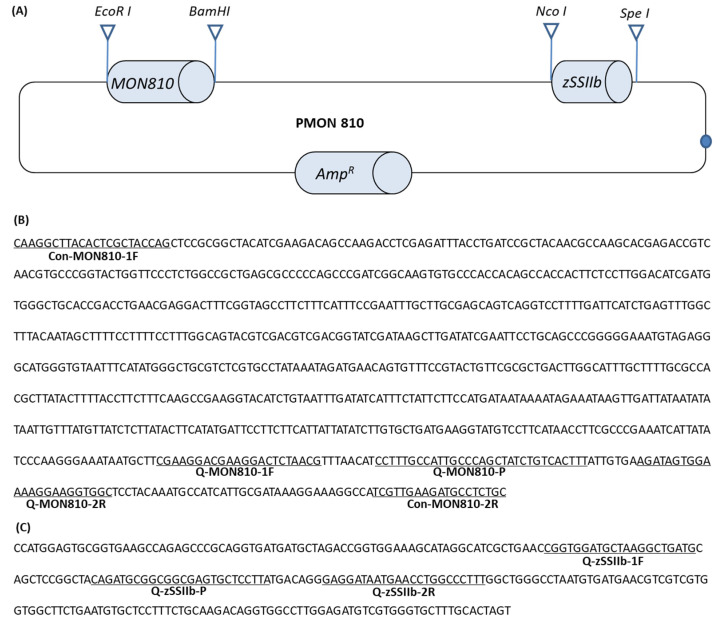
Structure and integration sequences of pMON810. (**A**) Schematic diagram of pMON810. *Amp^R^*, ampicillin-resistant gene; *MON810*, 5′ event-specific sequence of GM maize 810; *zSSIIb*, a specific fragment of maize endogenous reference gene *zSSIIb*; *EcoRI*, *Pst I*, *BamHI*, *NcoI* and *SpeI* indicate the corresponding restriction endonuclease sites. (**B**) Nucleotide sequences of the 5′ flanking sequence of GM maize MON810. (**C**) Partial nucleotide sequence of maize endogenous reference gene *zSSIIb*. The underlined and labeled sequences are the primer pairs and TaqMan probes used in this study.

**Figure 2 foods-11-01538-f002:**
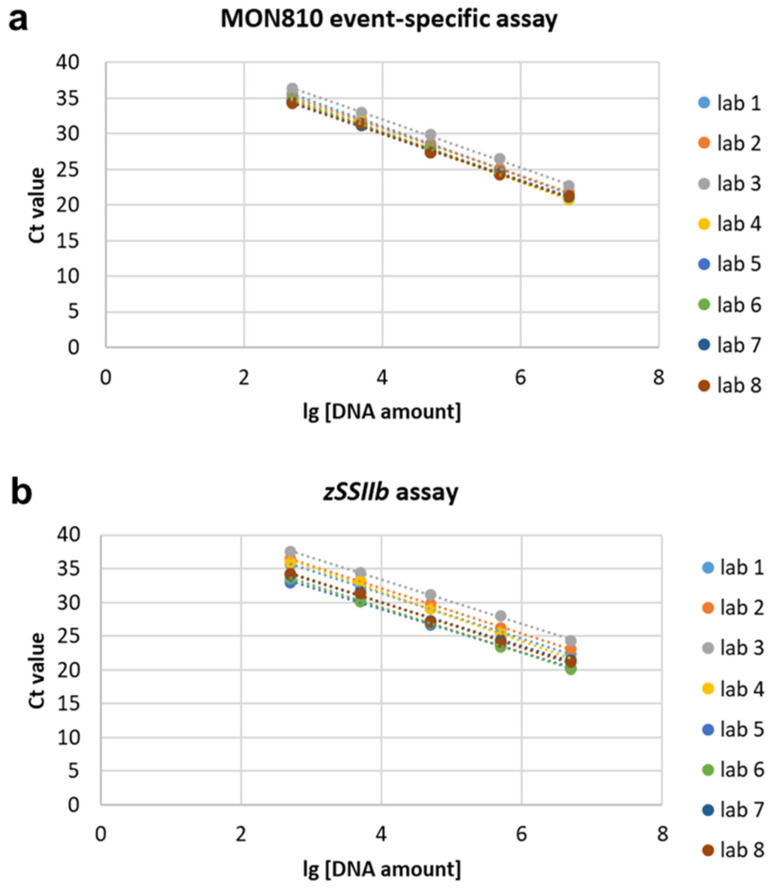
Constructed standard curves from qPCR assays employing pMON810 plasmid DNA dilutions as calibrators from eight participants. (**a**) The standard curves of MON810 assay. (**b**) The standard curves of *zSSIIb* assay.

**Figure 3 foods-11-01538-f003:**
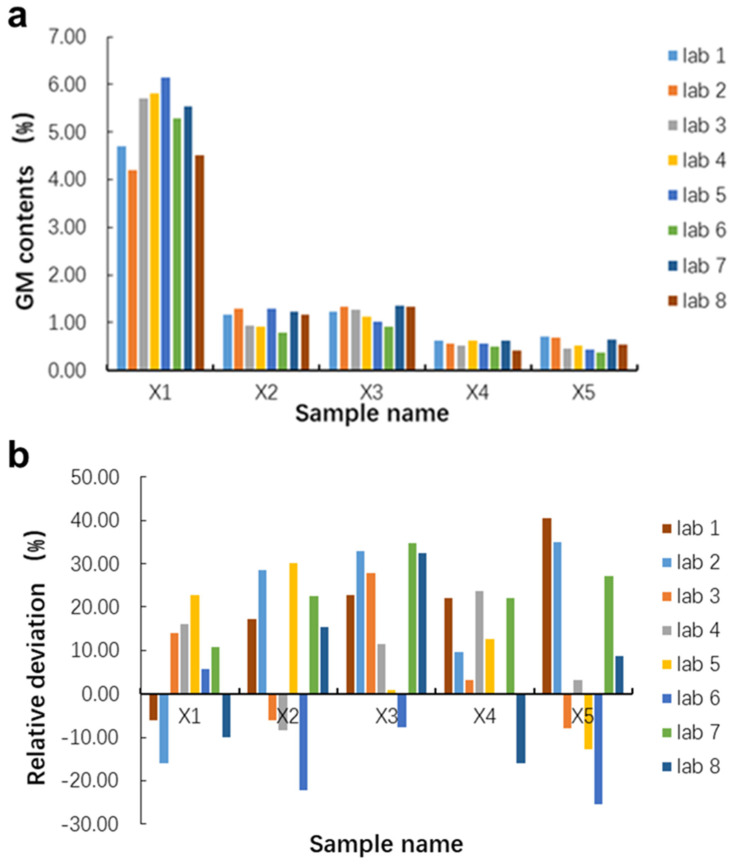
Relative deviation of the quantification results of five blind samples in the inter-laboratory study. (**a**) Quantified GM contents of blind samples in each participated lab. (**b**) The relative deviations of each blind sample in eight laboratories.

**Table 1 foods-11-01538-t001:** Formula, PCR efficiency, and linearity (R^2^) of each qPCR assay standard curve from eight laboratories.

Lab Code	MON810 Event-Specific								*zSSIIb*						Equipment
	Formula	Efficiency	SD	RSD (%)	R^2^	SD	RSD (%)	Formula	Efficiency	SD	RSD (%)	R^2^	SD	RSD (%)	
1	y = −3.4849x + 45.004	0.936	0.03	3.120	0.9997	0.001	0.093	y = −3.419x + 44.729	0.961	0.05	5.070	0.9995	0.002	0.216	Prism ABI 7500
2	y = −3.3641x + 44.305	0.983			0.9996			y = −3.3643x + 45.581	0.983			0.9997			Prism ABI 7500
3	y = −3.3803x + 45.607	0.976			0.9991			y = −3.3085x + 46.641	1.006			0.9989			Rotor-Gene Q
4	y = −3.5054x + 44.354	0.929			0.998			y = −3.6326x + 46.044	0.885			0.9971			CFX96 Touch
5	y = −3.3131x + 43.492	1.004			0.9992			y = −3.2092x + 41.843	1.049			0.9987			CFX96 Touch
6	y = −3.4103x + 43.977	0.964			0.9997			y = −3.3528x + 42.61	0.987			0.9996			Prism ABI 7900
7	y = −3.3181x + 43.275	1.002			0.9991			y = −3.244x + 43.019	1.034			0.9933			Prism ABI 7900
8	y = −3.3032x + 43.259	1.008			0.9971			y = −3.3571x + 43.372	0.986			0.9972			CFX96 Touch

**Table 2 foods-11-01538-t002:** Conversion factor (Cf) values from eight laboratories in the inter-laboratory study.

Lab Code	C1			C2			C3			C4			C5			Mean Cf	SD	RSD (%)
	Mean Ct		Cf	Mean Ct		Cf	Mean Ct		Cf	Mean Ct		Cf	Mean Ct		Cf			
	MON810	*zSSIIb*		MON810	*zSSIIb*		MON810	*zSSIIb*		MON810	*zSSIIb*		MON810	*zSSIIb*				
1	25.53	24.87	0.60	27.34	26.87	0.70	29.86	29.26	0.66	32.30	31.68	0.67	34.74	33.90	0.60	0.65	0.04	6.78
2	24.24	24.82	0.62	26.53	27.39	0.75	29.13	29.65	0.60	31.84	32.66	0.73	34.39	35.28	0.77	0.69	0.08	11.48
3	26.31	26.68	0.47	28.17	28.97	0.66	30.69	30.94	0.47	33.10	33.17	0.43	35.50	35.82	0.52	0.51	0.09	17.78
4	24.43	24.50	0.56	26.64	26.94	0.62	29.17	29.46	0.58	31.75	31.95	0.52	34.35	35.04	0.66	0.59	0.06	9.37
5	24.97	23.21	0.61	27.26	25.40	0.60	29.79	28.05	0.69	32.34	30.36	0.62	34.55	32.65	0.69	0.64	0.04	6.92
6	27.07	25.05	0.53	29.57	27.37	0.48	32.06	30.16	0.60	34.48	32.78	0.71	37.01	35.16	0.66	0.60	0.10	16.15
7	24.58	24.23	0.70	26.93	26.58	0.72	28.91	28.41	0.67	31.43	31.04	0.75	33.88	33.15	0.61	0.69	0.05	7.73
8	24.97	24.25	0.69	26.90	26.21	0.69	29.36	28.62	0.65	31.70	31.01	0.66	33.64	33.10	0.71	0.68	0.02	3.63

**Table 3 foods-11-01538-t003:** Quantification results of blind samples from eight laboratories.

Lab Code	qPCR Assay	X1			X2			X3			X4			X5		
		Ct	GM Content (%)	Mean	Ct	GM Content (%)	Mean	Ct	GM Content (%)	Mean	Ct	GM Content (%)	Mean	Ct	GM Content (%)	Mean
1	MON810	33.8	4.70	5.23	35.86	1.17	1.10	35.76	1.23	1.20	36.95	0.61	0.55	36.85	0.70	0.54
	*zSSIIb*	28.55			28.51			28.48			28.61			28.72		
2	MON810	33.05	4.20		34.86	1.29		34.72	1.33		35.91	0.55		35.57	0.68	
	*zSSIIb*	29.16			29.24			29.15			29.05			29.01		
3	MON810	33.25	5.70		35.61	0.94		35.26	1.28		36.43	0.52		36.82	0.46	
	*zSSIIb*	29.46			29.18			29.28			29.12			29.34		
4	MON810	31.46	5.80		34.55	0.92		34.28	1.11		35.06	0.62		35.27	0.52	
	*zSSIIb*	27.36			27.65			27.68			27.56			27.49		
5	MON810	31.65	6.14		33.49	1.30		33.68	1.01		34.28	0.56		34.72	0.44	
	*zSSIIb*	25.86			25.48			25.31			25.08			25.15		
6	MON810	32.05	5.28		34.52	0.78		34.86	0.92		35.27	0.50		35.61	0.37	
	*zSSIIb*	25.85			25.49			25.68			25.58			25.49		
7	MON810	31.57	5.54		33.62	1.23		33.34	1.35		34.45	0.61		34.29	0.64	
	*zSSIIb*	26.98			26.86			26.72			26.69			26.59		
8	MON810	31.54	4.50		33.68	1.15		33.59	1.33		35.05	0.42		34.82	0.54	
	*zSSIIb*	26.38			26.57			26.68			26.49			26.63		

**Table 4 foods-11-01538-t004:** Statistical analysis of the quantification results for the blind sample detection from eight laboratories.

Unknown Samples GM%	Blind Sample				
	X1	X2	X3	X4	X5
Laboratories returning results	8	8	8	8	8
Sample per laboratory	1	1	1	1	1
Total data number	9	9	9	9	9
Data excluded	0	0	0	0	0
Reason for exclusion	-	-	-	-	-
Mean value (%)	5.23	1.10	1.20	0.55	0.54
True value (%)	5.00	1.00	1.00	0.50	0.50
Repeatability SD	0.35	0.11	0.13	0.06	0.09
Repeatability RSD (%)	6.69	10.03	11.14	10.94	16.58
Reproducibility SD	0.69	0.19	0.16	0.07	0.12
Reproducibility RSD (%)	13.20	17.61	13.48	12.49	21.94
Bias (absolute value)	0.23%	0.10%	0.20%	0.05%	0.04%
Bias (%)	4.60	10.00	20.00	10.00	8.00

**Table 5 foods-11-01538-t005:** Evaluation of the laboratory variation and GM amounts in an inter-laboratory study using two-way analysis of variance (α = 0.05).

Source of Variation	Sum of Squares	Degrees of Freedom	Mean Squares	F	*p*
Error	7246.24	28	258.80		
Laboratory	3802.60	7	543.23	2.10	0.08
GM amounts	957.00	4	239.25	0.92	0.46
Total	12,005.83	39			

## Data Availability

The data presented in this study are available in the article and Appendix A.

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
