# Peer review of "Collaborative Ring Trial of the Applicability of a Reference Plasmid DNA Calibrant in the Quantitative Analysis of GM Maize Event MON810"

_foods, 2022, doi:10.3390/foods11111538_

Round 1

Reviewer 1 Report

In the manuscript entitled "Collaborative ring trial of the applicability of a reference plasmid DNA calibrant in the quantitative analysis of GM maize event MON810" authors constructed a plasmid DNA reference material, pMON810, for qPCR quantitative analysis of GM maize MON810, and they organized a collaborative ring trial for interlaboratory validation of the applicability and feasibility of pMON810 as a plasmid calibrant.

Why were the kernels of MON810 maize selected for research? Is this maize most popular in China or worldwide? Were the studies in cooperating laboratories carried out on identical devices? Is it possible to talk about the repeatability and consistency of the results, as well as their averaging, in the case of using different equipment, especially in relation to Table 4?

Mean value (%) 5.23 1.10 1.20 0.55 0.54 True value (%) 5.00 1.00 1.00 0.50 0.50 Similarly, Figure 3 requires a comment.

Why was it not specified which devices were used for measurements in individual laboratories?

This should be completed. The summary is very general and does not convince about the significant scientific value of this work.

Author Response

Response to comments from Referee 1

In the manuscript entitled "Collaborative ring trial of the applicability of a reference plasmid DNA calibrant in the quantitative analysis of GM maize event MON810" authors constructed a plasmid DNA reference material, pMON810, for qPCR quantitative analysis of GM maize MON810, and they organized a collaborative ring trial for interlaboratory validation of the applicability and feasibility of pMON810 as a plasmid calibrant.

Comment 1: Why were the kernels of MON810 maize selected for research? Is this maize most popular in China or worldwide?

Answer: Thanks for your comment. The MON810 was the most important GM mazed which were approved for planting or commercialization worldwide. In China, Mon810 maize was approved for using as raw material to produce foods and feeds. In addition, the MON810 event was found to be planted illegally in some places in China. According to Chinese GMO labelling regulation, the food or feed products containing the GM content must be labelled. Therefore, we developed the plasmid DNA RM targeting MON810 maize event.

Comment 2: Were the studies in cooperating laboratories carried out on identical devices?

Answer: Thanks for your comment. In this ring trial, the identical qPCR equipment was not requested. Five different qPCR equipment was used among the eight labs, such as Rotor-Gene Q (Qiagen, Germany), Prism ABI 7300, Prism ABI 7500, Prism ABI 7900(Applied Biosystems, USA), and CFX96 Touch (BioRad, USA). (Line 131-134)

Comment 3: Is it possible to talk about the repeatability and consistency of the results, as well as their averaging, in the case of using different equipment, especially in relation to Table 4?

Answer: Thanks for your comment. We have added more details about the repeatability and consistency of the results. The details were shown in Figure 3. The SD and RSD values were also calculated to evaluate the consistency of the results from all eight laboratories. (Line 254-259)

Comment 4: Mean value (%) 5.23 1.10 1.20 0.55 0.54 True value (%) 5.00 1.00 1.00 0.50 0.50 Similarly, Figure 3 requires a comment.

Answer: Thanks for your comment. We have explained the results of Figure 3 in the revised MS. (Line 249-254)

The mean GM contents values in the five blind samples were 5.23%, 1.10%, 1.20%, 0.55% and 0.59%, respectively. The relative deviations (persent) of the results from the true value were acceptable except two data at 0.5% GM content (As shown in Figure 3.

Answer: Thanks for your comment. We have noted these two data, and described them in the MS. Also, we have discussed it. (Line 249-254)

Comment 5: Why was it not specified which devices were used for measurements in individual laboratories?

Answer: Thanks for your comment. In this work, we aimed to develop one plasmid DNA RM for the detection of GM maize MON810 in qPCR analysis. To prove whether it has wide applicability and insensitivity to equipment, we do not limit the test to specific types of equipment. Also, it is inappropriate to require the use of specific equipment in practical sample analysis in various labs. In this ring trial, five different qPCR equipment was used among the eight labs, such as Rotor-Gene Q (Qiagen, Germany), Prism ABI 7300, Prism ABI 7500, Prism ABI 7900(Applied Biosystems, USA), and CFX96 Touch (BioRad, USA). (Line 131-134). The ideal and consistent results were obtained from various equipment, indicating the pMON810 has wild applicability in further GMO analysis.

Comment 6: This should be completed. The summary is very general and does not convince about the significant scientific value of this work.

Answer: Thanks for your comment. We have revised the conclusion and added its potential application of pMON810.

Reviewer 2 Report

The manuscript presents the results obtained in the collaborative ring trial for the applicability of pMON810 in quantification of MON810 GM event. Overall, the research design is appropriate and the results are correct.

  • L85-92: Adjutst text size.
  • Figure 1. (b) and (c): Consider to improve the presentation of nucleotide sequences. 
  • Figure 2. Separate the two graphs in different figures or use (a) and (b) to describe them in the caption.
  • Figure 3: Same than Figure 2.
  • Avoid large blank espaces in text pages. 
  • English language needs improvement

Author Response

Response to comments from Referee 2

The manuscript presents the results obtained in the collaborative ring trial for the applicability of pMON810 in quantification of MON810 GM event. Overall, the research design is appropriate and the results are correct.

Comment 1: L85-92: Adjutst text size.

Answer: Thanks for your comment. We have revised in the new MS.

Comment 2: Figure 1. (b) and (c): Consider to improve the presentation of nucleotide sequences.

Answer: Thanks for your comment. The Figure 2b and 2c showed the sequences of target genes and used primers/probes in this work. The current presentation displayed this information clearly. Therefore, we didn't modify the graph further in the new MS.

Comment 3: Figure 2. Separate the two graphs in different figures or use (a) and (b) to describe them in the caption.

Answer: Thanks for your comment. We have revised it in the new MS.

Comment 4: Figure 3: Same than Figure 2.

Answer: Thanks for your comment. We have revised it in the new MS.

Comment 5: Avoid large blank espaces in text pages.

Answer: Thanks for your comment. We have revised it in the new MS.

Comment 6: English language needs improvement

Answer: Thanks for your comment. One native English speaker helped us to improve the language, all the modifications were highlighted with red color in the new MS.

Round 2

Reviewer 1 Report

The devices on which individual laboratories conducted their research were still not included in the manuscript. As a rule, the equipment used is mentioned in the manuscript. In order to compare the results from different laboratories, the experimental procedures should be identical. Thus the sentence: "The larger relative deviation came from the blind sample X5 with 0.5% GM content in two laboratories, which might be caused by the different DNA extraction methods used in the participated laboratories and the quantitative accuracy of the qPCR assays decreased in low concentration samples qPCR assays. (line 248-251) "shows that the methods were different, not identical. So how can these results be compared? Can such experiments be compared? The present summary is only shorter than the first version and it has made nothing significant. It is even more general in nature, and I have asked for elaboration and specification. The scientific value of the manuscript is still not increased and I suggest publishing the manuscript in a regional journal, especially as it concerns maize grown in China.

Author Response

Comment from Referee 1

The devices on which individual laboratories conducted their research were still not included in the manuscript. As a rule, the equipment used is mentioned in the manuscript. In order to compare the results from different laboratories, the experimental procedures should be identical. Thus the sentence: "The larger relative deviation came from the blind sample X5 with 0.5% GM content in two laboratories, which might be caused by the different DNA extraction methods used in the participated laboratories and the quantitative accuracy of the qPCR assays decreased in low concentration samples qPCR assays. (line 248-251) "shows that the methods were different, not identical. So how can these results be compared? Can such experiments be compared? The present summary is only shorter than the first version and it has made nothing significant. It is even more general in nature, and I have asked for elaboration and specification. The scientific value of the manuscript is still not increased and I suggest publishing the manuscript in a regional journal, especially as it concerns maize grown in China.

Answer: Thanks for your kindly comment. We have added the used qPCR equipment in each participant in Table 1. During the collaborative ring trial, we supplied the identical protocol to each participant, and all participants performed the validation according to the protocol.

Considering various DNA extraction methods or commercial kits and equipment are often used in different labs in routine analysis, we did not require all the labs to extract the DNA with same methods or kits and equipment in the ring trial. Although this design makes it much harder to verify experimentally, we believe that this design can make our experimental verification closer to the actual application scenario, which can be better prove the applicability of our constructed plasmid pMON810 in GMO quantification.

The validated results indicated that all the labs obtained the reliable results in blind samples quantification, although the eight labs extracted the DNAs using different methods. The results of verification just better prove the applicability of pMON810 RM in GMO quantification.

In the sections of “Abstract” and “Introduction”, we emphasized the importance of MON810 event detection for legal planting and using worldwide, and our work can supply an important plasmid RM for the detection and quantification of MON810 maize event which is helpful for all the countries and regions in GM maize commercial planting and production.

We do hope our reply will answer your questions well. Thanks again.

Reviewer 2 Report

The reviewer's previous concerns have been addressed with a detailed point-by-point response and the revised version of the manuscript has been improved accordingly. 

Author Response

Thanks for your confirmation and help on our work.